# Dynamic Landscape of Extracellular Vesicle-Associated Proteins Is Related to Treatment Response of Patients with Metastatic Breast Cancer

**DOI:** 10.3390/membranes11110880

**Published:** 2021-11-16

**Authors:** Olivia Ruhen, Xinyu Qu, M. Fairuz B. Jamaluddin, Carlos Salomon, Aesha Gandhi, Michael Millward, Brett Nixon, Matthew D. Dun, Katie Meehan

**Affiliations:** 1Sarcoma Molecular Pathology Team, Institute of Cancer Research, Sutton SW7 3RP, UK; olivia.ruhen@icr.ac.uk; 2Department of Otorhinolaryngology, Head and Neck Surgery, Chinese University of Hong Kong, Hong Kong; xyqu@surgery.cuhk.edu.hk; 3School of Biomedical Sciences and Pharmacy, College of Health, Medicine and Wellbeing, University of Newcastle, Callaghan, NSW 2308, Australia; muhammad.jamaluddin@newcastle.edu.au (M.F.B.J.); Matt.Dun@newcastle.edu.au (M.D.D.); 4Hunter Medical Research Institute, Cancer Research Program, New Lambton Heights, NSW 2305, Australia; 5Exosome Biology Laboratory, Centre for Clinical Diagnostics, UQ Centre for Clinical Research, Royal Brisbane and Women’s Hospital, University of Queensland, Brisbane, QLD 4072, Australia; c.salomongallo@uq.edu.au; 6Departamento de Investigación, Postgrado y Educación Continua (DIPEC), Facultad de Ciencias de la Salud, Universidad del Alba, Santiago 8320000, Chile; 7Fiona Stanley Hospital, Perth, WA 6009, Australia; Aesha.Gandhi@health.wa.gov.au; 8School of Medicine, University of Western Australia, Perth, WA 6009, Australia; Michael.millward@uwa.edu.au; 9Reproductive Science Group, College of Engineering, Science and Environment, University of Newcastle, Newcastle, NSW 2308, Australia; Brett.Nixon@newcastle.edu.au

**Keywords:** metastatic breast cancer, extracellular vesicles, treatment response, quantitative proteomics

## Abstract

Breast cancer is the leading cause of cancer death in women. The majority of these deaths are due to disease metastasis, in which cancer cells disseminate to multiple organs and disrupt vital physiological functions. It is widely accepted that breast cancer cells secrete extracellular vesicles (EVs), which contain dynamic molecular cargo that act as versatile mediators of intercellular communication. Therefore, Evs. secreted by breast cancer cells could be involved in the development of metastatic disease and resistance to treatment. Moreover, changes in EV cargo could reflect the effects of therapy on their parent tumor cells. The aim of this feasibility study was to quantitatively profile the proteomes of Evs. isolated from blood samples taken from treatment sensitive and resistant metastatic breast cancer patients to identify proteins associated with responses. Three serial blood samples were collected from three patients with metastatic breast cancer receiving systemic therapy including a responder, a non-responder, and a mixed-responder. Evs. were isolated from plasma using size exclusion chromatography and their protein cargo was prepared for tandem mass tag (TMT)-labelling and quantitative analyses using two-dimensional high-performance liquid chromatography followed by tandem mass spectrometry. After filtering, we quantitatively identified 286 proteins with high confidence using a q value of 0.05. Of these, 149 were classified as EV associated candidate proteins and 137 as classical, high abundant plasma proteins. After comparing EV protein abundance between the responder and non-responder, we identified 35 proteins with unique de-regulated abundance patterns that was conserved at multiple time points. We propose that this proof-of-concept approach can be used to identify proteins which have potential as predictors of metastatic breast cancer response to treatment.

## 1. Introduction

Breast cancer is the most commonly diagnosed cancer and leading cause of cancer death in women [1]. This incidence is predicted to increase over the next decade, particularly in developed nations where there is a prevalence of breast cancer risk factors attributed to lifestyle [2,3]. Most cases of breast cancer are detected in earlier stages due to screening. However, up to 30% of these patients will experience metastatic relapse [4]. While meaningful improvements in survival have been seen thanks to the introduction of newer systemic therapies, metastatic breast cancer remains a largely incurable disease [4]. Up to 90% of breast cancer-associated deaths are related to metastatic disease, which is thought to be a result of disrupted physiological function in multiple organs due cancer cell dissemination [5].

The development of metastases is a complex process with multiple theories on its mechanism. One such theory is the invasion-metastasis cascade, a multi-step process whereby there is dissemination of cancer cells from primary tumors to distant tissues to form new tumor colonies [6]. This theory involves local invasion of primary tumor cells to the surrounding tissues, intravasation of cells into the circulatory system and survival during hematogenous spread, arrest and extravasation through vascular walls into the parenchyma of distant tissues, formation of micrometastatic colonies, and subsequent proliferation of microscopic colonies into apparent clinically detectable lesions [7]. Metastatic colonization is largely influenced by the microenvironment of the secondary organ, with certain organs producing soluble components that support metastatic behavior. Additionally, the primary tumor has the potential to “prime” or augment distant organ microenvironments to aid metastases. A large variety of soluble factors secreted by the primary tumor initiates priming [8,9]. These molecules target the bone marrow for recruitment and initiation of remodeling of the target secondary organ, creating a premature metastatic niche. Bone marrow derived cells and other immune regulatory cells are recruited to the secondary site by continual secretion of factors from the primary site. The immune-suppressed environment so created is conducive for cancer colonization. Some disseminated cancer cells may stay in a dormant state until conditions of the secondary site are compatible with supporting tumor outgrowth into micrometastases. Thereafter, tumor-secreted factors regulate the growth of micrometastases to macrometastases during a progressive state [8,9,10].

One such tumor-derived factor thought to be involved in the metastatic cascade are extracellular vesicles (EVs) [11,12,13]. Evs. are a population of small membranous vesicles released from cells, which carry a diverse array of proteins, lipids and nucleic acids [14]. Evs. can be transported to distant sites through the hemolymphatic system, where they can bring about physiological changes to recipient cells via unloading of their molecular cargo. In this manner, Evs. are known to play a significant role in tumor growth and progression, with several studies implicating EV-associated molecules in the process of metastasis [8,9]. In addition, Evs. have been linked to the resistance of breast cancer cells to treatment, which is another mechanism by which they can promote progression to metastatic disease [15]. As such, it stands to reason that changes in the EV cargo over time could reflect the response of tumors to therapy and hence investigations to characterize their molecular cargo could help to elucidate the mechanisms by which metastatic progression occurs. Accordingly, the aim of this proof-of-principle study was to quantitatively profile the proteomes of Evs. isolated from the plasma of three women with metastatic breast cancer throughout their treatment to identify EV-associated proteins that may be associated with therapeutic response.

## 2. Materials and Methods

### 2.1. Cohort

Participants with metastatic breast cancer were recruited at Sir Charles Gairdner Hospital, WA following ethical approval from the Human Research Ethics Committee (reference numbers SCGH HREC 2013-051 and 2013-222). Study participants provided written informed consent prior to study enrolment. Diagnoses were pathologically confirmed. Serial blood samples (20 mL) were collected at QEII, Pathwest laboratories from each patient at three different time points yielding in a total of nine plasma samples included for analysis. All samples were processed within 2 h of collection.

### 2.2. Extracellular Vesical Isolation

Whole blood from breast cancer patients was collected in EDTA Vacutainer^®^ blood collection tubes (Becton Dickinson, Sydney, NSW, Australia) and processed within 2 h of collection. First, blood was centrifuged at 1600× *g* for 20 min. The plasma fraction was further clarified by centrifugation at 16,000× *g* for 10 min. Plasma (1 mL) was clarified once again by centrifugation at 10,000× *g* for 10 min and then diluted 1 in 4 with filtered PBS prior to loading onto a rinsed qEV column (Izon, Christchurch, New Zealand). Thirteen elution fractions of 500 µL were collected. Fractions 9 to 12 were pooled based on the presence of common EV markers CD9 and TSG101, as determined by Western blot. Ultracentrifugation was performed as per our previous studies [1,2].

### 2.3. Transmission Electron Microscopy

Evs. resuspended in PBS were fixed in 2% paraformaldehyde and transferred onto 200 mesh Formvar-carbon coated copper grids (ProSciTech, Brisbane, QLD, Australia). Samples were adsorbed for 20 min at RT prior to being washed twice in filtered PBS and four times in 50 mM glycine to quench free aldehyde groups. Samples were then blocked in 5% bovine serum albumin (BSA, Sigma, Sydney, NSW, Australia) for 10 min before being incubated with primary antibody solution (mouse anti-human CD9, clone MM2/57; 10 μg/mL, Merck, Perth, WA, Australia) for 30 min. Grids were then washed four times in 0.1% BSA and four times in 0.5% BSA before incubation with secondary antibody (goat anti-mouse IgG-gold conjugate, 1:24, Aurion, Wageningen, The Netherlands) for 20 min. Labelled samples were washed six times in PBS, fixed in 1% glutaraldehyde and washed six times in deionised water before counterstaining with 1% uranyl acetate. Following 2 min of counterstaining, grids were left overnight to dry. Grids were visualised on the JEOL JEM-2100 electron microscope (JEOL, Tokyo, Japan) at an operating voltage of 120 kV. Images were captured using an 11-megapixel Gatan Orius digital camera (Gatan, CA, USA).

### 2.4. Nanoparticle Tracking Analysis (NTA)

EV samples (100 μL) were diluted 1/1000 in PBS and injected into the analysis chamber of the NanoSight NS500 Instrument (Malvern Panalytical, Sydney, NSW, Australia) for particle size and concentration analysis. This instrument is equipped with a 405 nm laser and a sCMOS camera to detect the Brownian motion of light-scattering particles as they move through the solution. Sample analysis was performed at a camera level of 10 and gain of 250, with a detection threshold of 10 pixels. Settings for blur, minimum track length and minimum expected size were set to ‘auto’. Videos were recorded for 60 s at 30 frames/second in triplicate at 25 °C. All post-acquisition settings remained constant between samples. NTA software v3.0 was used to process and analyse the data. Each video was analysed to generate particle size (nm) distribution profiles and concentration values (particles/mL solution), which were downloaded as a report with the results of quality control analysis. The raw observational data was exported into Microsoft^®^ Excel as a comma-separated values (CSV) file.

### 2.5. Western Blot

EV samples in 100 μL PBS were mixed with 100 μL of RIPA buffer (Sigma-Aldrich, Sydney, NSW, Australia) and protease inhibitors (Roche Diagnostics, Sydney, NSW, Australia) and incubated on ice for 1 h to lyse the vesicle membranes. EV proteins were collected by centrifugation at 16,000× *g* for 15 min. The EV proteins were diluted in 4× Laemmli Buffer (Bio-Rad, Sydney, NSW, Australia) for a final loading volume of 20 μL. Proteins were separated at 155V for 30 min on a Mini Protean^®^ TGX 5–15% Stain-Free™ Precast Gel (Bio-Rad) using the Mini Protean^®^ Tetra Cell (Bio-Rad, Sydney, NSW, Australia). Proteins were then transferred onto a Trans-Blot^®^ Turbo™ Mini Nitrocellulose Membrane (Bio-Rad, Sydney, NSW, Australia) using the Trans-Blot^®^ Turbo™ Transfer System (Bio-Rad, Sydney, NSW, Australia) at a constant voltage of 25V. The membrane was then probed for EV proteins using the iBind™ Flex Western Device (Thermo Fisher Scientific, Sydney, NSW, Australia), commencing with blocking for 20 min in iBind™ Flex Solution (Thermo Fisher Scientific, Sydney, NSW, Australia). Primary antibody solutions of mouse anti-human CD9 (clone MM2/57, Merck, Perth, WA, Australia), mouse anti-human Calnexin (clone 1563, Novus Biologicals, CO, USA), and rabbit anti-human TSG101 (clone EPR7130B, Abcam, Melbourne, VIC, Australia) were created, with dilutions of 1:250 and 1:500 for anti-CD9 and Calnexin/TSG101, respectively. Samples were incubated with primary antibody solution before being incubated with secondary antibodies (sheep anti-mouse IgG-HRP conjugate, 1:2000, GE Healthcare, Sydney, NSW, Australia; goat anti-rabbit IgG-HRP conjugate, 1:2000, Merck, Perth, WA, Australia) for a combined incubation time of 2.5 h. Signals were developed using Clarity™ Western ECL Blotting Substrates (Bio-Rad, Sydney, NSW, Australia) and were subsequently imaged using the ChemiDoc™ Touch Imaging System (Bio-Rad, Sydney, NSW, Australia). Images were processed using Image Lab™ software v6.0 (Bio-Rad, Sydney, NSW, Australia).

### 2.6. Extracellular Vesicle Protein Digestion and Labelling

EV protein preparations from three respective cases of metastatic breast cancer including a responder (*n* = 3 patients), a non-responder (*n* = 3), and a mixed responder (*n* = 3) were generated for analysis in this study. The proteins were precipitated with 4 volumes of cold acetone and incubated overnight at −20 °C. Precipitated proteins were centrifuged at 14,000× *g* for 15 min at 4 °C. Air dried protein pellets were prepared for MS by dissolving in 6 M urea and 2 M thiourea containing protease inhibitor cocktail (Sigma, Sydney, NSW, Australia) and phosphatase inhibitors (Roche, Sydney, NSW, Australia) [3]. The cysteine residues were reduced by adding 10 mM dithiothreitol (DTT) for 1 h at room temperature, and alkylated by adding 20 mM iodoacetamide (Sigma) for 1 h in the dark at room temperature. Proteins were then cleaved using 1:25 ratio Lys-C/trypsin (Promega) and incubated for 3 h at room temperature. The concentration of urea was then reduced below 0.75 M by adding 50 mM triethylammonium bicarbonate (TEAB, Sigma, Sydney, NSW, Australia), pH 7.8 and digested overnight at room temperature [4]. Lipid precipitation was achieved by centrifuging the acidified samples at 14,000× *g* for 10 min at room temperature, and further processed for trichloroacetic acid (TCA, Sigma, Sydney, NSW, Australia) precipitation. TCA precipitation of the remaining lipopeptides were then recombined with supernatant containing peptides before deglycosylation with PNGase F (New England Biolabs, MA, USA) and Glyko^®^ Sialidase A (Agilent, Melbourne, Vic, Australia) [5]. Deglycosylation was performed overnight at 37 °C. Peptides were cleaned up using a modified StageTip microcolumn and solid phase extraction (SPE) columns (Oasis PRIME HLB, Waters, Rydalmere, NSW, Australia) [6]. Peptide concentrations were determined using a PierceTM quantitative fluorometric peptide assay (Thermo Fisher Scientific, Sydney, NSW, Australia). Peptides were dried by SpeedVac, and reconstituted in 50 mM TEAB pH 8. Reconstituted peptides were then processed according to the manufacturer’s protocol for tandem mass tag (TMT) 10 plex (Thermo Fisher Scientific, Sydney, NSW, Australia) for comparative and quantitative analyses (TMT 10 plex labels; mixed response patient time point 1 = 126, mixed response patient time point 2 = 127 N, mixed response patient time point 3 = 127 C, responder patient time point 1 = 128N, responder patient time point 2 = 128 C, responder patient time point 3 = 129 N, non-responder patient time point 1 = 129 C, non-responder patient time point 2 = 130 N, non-responder patient time point 3 = 130 C) [7]. Digestion and TMT labelling efficiency was determined by LC-MS/MS. Samples were then mixed in 1:1 ratio and fractionated by hydrophilic interaction chromatography (HILIC) using a Dionex UltiMate 3000 capLC system (Dionex, California, USA) prior to nanoLC-MS/MS.

### 2.7. Liquid Chromatography Tandem Mass Spectrometry Analysis

Hydrophilic interaction chromatography (HILIC) was performed on a Dionex UltiMate 3000 capLC system using a TSKgel Amide-80 HILIC columns packed with 3 μm particles (4.6 mm ID × 15 cm) (Tosoh Biosciences, Pennsylvania, USA) connected in line, with the following mobile phases: 0.1% TFA in HPLC water (solvent A) and 0.1% TFA in acetonitrile (Acn) (solvent B). Peptides were separated at a flow rate of 6 µL/min with a 35-min linear gradient from 98% to 25% solvent B. 14 fractions were collected into a deep 96-well LoBind plate (Eppendorf, Sydney, NSW, Australia), dried, and resuspended in 2% Acn containing 0.1% TFA [3,4,6]. HILIC fractionated peptides were then subjected to nanoflow LC instrument interfaced to a Q Exactive Plus Orbitrap mass spectrometer (Thermo Fisher Scientific, MA, USA), which is equipped with a nanoelectrospray ion source. Peptides were loaded onto a trapping column (Acclaim PepMapTM100, 75 µm × 2 cm, nanoViper fitting C18, 3 µm, 100 Å, Thermo Fisher Scientific, Sydney, NSW, Australia) for pre-concentration, and then resolved in the analytical column (EASY-Spray Column, PepMap, 75 µm × 15 cm, nanoviper fitting, C18, 2 µm, 100 Å, Thermo Fisher Scientific, MA, USA) for peptide separation. The separation of the peptides were achieved at a constant flow rate of 250 nL/min, using a linear gradient from 5% to 40% of solvent B (95% Acn, 0.1% TFA) over 95 min. Ions were generated by positive electrospray ionization via liquid junction into a Q Exactive mass spectrometer. Mass spectra were acquired over *m*/*z* 350–2000 at 70,000 resolution and data-dependent acquisition selected the top fifteen most abundant precursor ions for tandem mass spectrometry by HCD fragmentation, collision energy of 32.0, and a resolution of 35,000. A 1.4 *m*/*z* isolation window and a fixed first mass of 120 *m*/*z* was used for MS/MS scans. Automatic gain control targets were 3E6 ions for Orbitrap scans and 5E5 for MS/MS scans. Dynamic exclusion was set at 15 s, as well as rejection of precursor ions with charge state +1, +7 and greater than +8, were employed to minimize redundant MS/MS collection and maximize peptide identifications [3].

### 2.8. Data Analysis

The MS raw files were processed using Proteome Discoverer (PD) software (version 2.1, Thermo Fisher Scientific, Sydney, NSW, Australia). In the PD analysis, MS/MS spectra were search against the UniProt human protein database (73,653 sequences, updated 16 January 2019). The mass tolerance of precursor mass and fragment mass were set as 10 ppm and 0.02 Da respectively. S-carbamidomethylation of cysteine was set as a fixed modification; methionine oxidation, lysine acetylation, asparagine or glutamine deamidation and TMT labelling of amines and lysine were set as variable modifications. For digestion, trypsin was set as the digestion enzyme with two missed cleavage permitted. A fixed false discovery rate (FDR) was set at <1% for identification of peptides and proteins. Protein localisation and functions were interrogated using FunRich (version 3.1.3) and String (version 11.0) [8,9]. Principle components analysis and hierarchical clustering were performed using ClustVis [10].

## 3. Results

### 3.1. Patient and Clinical Characteristics

Evs. were isolated from the blood of three patients with metastatic breast cancer at three time points during treatment. Table 1 shows the clinical characteristics of the patients included in this study. Computerised tomography (CT) and nuclear medicine (NM) scans demonstrated the patients’ disease status (responding to treatment, stable disease or progressive disease as defined by RECIST 1.1 guidelines) at certain time points during treatment. Patients were classified as either a responder, non-responder or mixed responder according to their overall response to treatment during the study period. The time points of blood collection, scans, and treatment are shown in Figure 1. The responder was a 65-year-old female with stage IV ER+ and PR+ invasive ductal carcinoma (IDC). A CT scan performed in January 2016 indicated new metastatic disease after the patient had received a combination of adjuvant tamoxifen and capecitabine, which commenced in February 2016. A CT/NM scan in August 2016 and February 2017 showed stable disease and reduced metastases without new lesions, which suggested the patient had responded to the treatment. The non-responder was a 68-year-old female with stage IV ER+ and PR+ IDC, initially treated with anastrazole. A NM scan in February 2016 indicated stable bone metastasis but later in July of the same year, the metastases had expanded. Therefore, due to disease progression and rising cancer antigen 15.3 (a traditional serum marker for breast cancer monitoring), the treatment was changed to exemestane and everolimus. Unfortunately, CT/NM scans in both December 2016 and March 2017 showed an increase in size of primary tumour and enlarged nodules, which suggested the tumour had progressed further. The mixed responder was a 69-year-old female with stage IV ER+ and PR+ IDC. After three cycles of chemotherapy of capecitabine alone, a CT scan in June 2016 showed progressive metastatic disease. The patient received radiotherapy and a second CT scan was performed in September 2016, which revealed no residual disease within the thorax or abdomen. Bony metastases appeared generally more sclerotic and chemotherapy with gemcitabine and abraxane was commenced in December 2016. CT/NM scans in December 2016 showed progressive disease, illustrating that at the first time point of blood collection the patient had responded to the chemotherapy but then failed to respond to subsequent treatments.

### 3.2. Validation of Successful Extracellular Vesicle Isolation

Plasma borne Evs. were enriched using size exclusion chromatography (SEC) from blood samples (Materials and Methods) of three patients with metastatic breast cancer at three time points during their treatment (Figure 1). Evs. were subjected to nanoparticle tracking analysis (NTA) revealing the mean size of isolated Evs. was approximately 146 nm (range 60–540 nm) and the concentration was approximately 3.11 × 10^10^ particles/mL of plasma (range 2.45 × 10^10^–6.83 × 10^10^ particles/mL plasma, Figure 2A). This is consistent with previous reports showing that the size and concentration of plasma Evs. from patients with breast cancer ranges from 80–160 nm and 6.00 × 10^9^–8.05 × 10^11^ particles/mL respectively [11,12,13,14]. Transmission electron microscopy (TEM) analysis revealed spherical particles with a bilayer membrane in the size range of 50–150 nm consistent with previous reports (Figure 2B) [11,12,13,14]. Western blot analysis detected EV protein biomarkers according to internationally accepted guidelines (Figure 2C, Appendix A) [15]. CD9, a known EV marker involved in EV biogenesis, maturation, and secretion, was detected regardless of the isolation method. In contrast TSG101, another classical EV marker, was only detected in the pooled SEC EV preparation. Though these proteins were detected by western blot, they and some other classic EV markers were not detected by mass spectrometry. This may be due to the highly hydrophobic nature of tetraspanins and large membrane proteins, a feature that renders them difficult to analyse by mass spectrometry [16]. On the contrary, calnexin, an endoplasmic reticular protein, was not detected in the EV samples by western blot or mass spectrometry, proving that the isolated Evs. were purified and not contaminated by redundant intracellular components. In concordance with previous reports, we were unable to routinely identify certain tetraspanins (CD63/81) in plasma Evs. [17]. This provides additional evidence that tetraspanins are not universal markers for Evs. isolated from plasma. Taken together, these results demonstrated that Evs. were successfully isolated from the plasma with high purity and well characterized by various methods.

### 3.3. Quantitative Extracellular Vesical Proteomic Profiling

Isolated Evs. (Figure 2) were subjected to tryptic digestion [5], tandem mass tagging (TMT) [4] and quantitative proteomics [3,18] which identified 470 unique proteins, with 113 excluded based on a q value which is greater than 0.05 and considered to be an insignificant change in expression, 67 excluded due to missing tags and incomplete quantitative data, and 4 excluded due to non-human origin (Appendix A). For the remaining 286 proteins, raw abundance values of each protein were converted into a ratio by dividing reporter-ion abundance values at various time points (1/2, 1/3, and 2/3). Proteins with an empirical fold change of greater than 1.5 were considered de-regulated. Both high-abundant plasma proteins (such as albumin, fibrinogen, haptoglobin, alpha-2-macroglobulin, and various immunoglobulins) and classical EV proteins (such as heat shock protein HSP 90-beta, tubulin, and von Willebrand factor) were identified. Albumin and immunoglobulins are two of the most abundant plasma proteins, and their detection not only implies plasma protein contamination, but may also have reduced the sensitivity of the mass spectrometry (MS) by narrowing its dynamic range of analysis. On the other hand, heat shock protein HSP 90-beta, tubulin and von Willebrand factor were used as indicators of plasma EV enrichment. tubulin is a cytoskeleton protein while heat shock protein HSP 90-beta is a cytosolic protein and both have been reported to be enriched in Evs. [19,20,21]. von Willebrand factor is one of the most common protein markers found in platelet-derived Evs. [19,20], which accounts for up to 90% of Evs. found in plasma. Due to the classical, high abundant, plasma protein nature of 137 proteins, we have analysed these separately in order to maximise the capacity of the analysis. The remaining 149 EV associated proteins of interest were compared within and between the responder and non-responder (Appendix A) and considered in the mixed-responder (Appendix A). The MS proteomic data were deposited in the Mass Spectrometry Interactive Virtual Environment (MassIVE) database with the dataset identifier: MSV000087527 and are publicly accessible by the following link: (https://massive.ucsd.edu/ProteoSAFe/dataset.jsp?task=4e6ad1a618d24d79b648d72af5a9e0d3).

### 3.4. Analysis of Extracellular Vesical Associated Proteins of Interest

EV proteins (*n* = 149) were categorized according to their GO terms to explore the enrichment of cellular functions. Initially, proteins were categorized according to the cellular compartment with which they associate (Figure 3A). Most proteins (>65%) were associated with exosomes and the extracellular space but some were also categorized as associated with the nucleosome, centrosome, lysosome or cytoskeleton. As nucleosome, centrosome and lysosome related proteins are unlikely to be present in EVs, it is possible that we have co-isolated extra-vesicular proteins. Nevertheless, the putative EV protein cargo were also categorized according to their role within biological pathways, a strategy that identified the complement cascade, initial triggering of complement, RNA processing, and telomere maintenance as the most overrepresented within these vesicles (Figure 3B).

Analysis of proteins showing altered abundance over time between the responder and non-responder (Table 2) identified 23 proteins that increased abundance in the non-responder; 7 proteins decreased in the non-responder only, and 5 proteins increased in the responder (Figure 4, Table 3, Table 4 and Table 5). Six proteins displayed a greater than five-fold change in abundance between any given time point, including: cofilin-1, tropomodulin-2, peroxiredoxin-2, vimentin, histone H2B type 1-K and multimerin-1. Collectively, these 35 proteins were able to discriminate the responder from the non-responder at any given time point by supervised principal components analyses and hierarchical clustering (Figure 5). The principal components analysis also revealed that the proteomic profile of the mixed responder grouped more closely with that of the non-responder which is consistent with the ultimate lack of response observed in this case. According to the human protein atlas, over 70% of these proteins (25 out of 35) are expressed in breast tumours based on immunohistochemistry. Additional in silico analysis of these de-regulated proteins revealed that high breast tumoral expression of fibronectin, fructose-bisphosphate aldolase A, pyruvate kinase PKM, peptidyl-prolyl cis-trans isomerase A, histone H3.1, histone H2A type 2-A, alstrom syndrome protein 1, fibulin-1, titin, glyceraldehyde-3-phosphate dehydrogenase, tropomodulin-2, cofilin-1, and peroxiredoxin-2 were significantly associated with shorter overall survival (Figure 6) [22]. In contrast, low tumoral expression of Protein S100-A6 and Plasma kallikrein were associated with shorter overall survival.

### 3.5. Extracellular Vesical Associated Proteins Involved in the Complement Cascade

Considering previous evidence that Evs. contain key complement factors and regulators [23], we examined proteins in this cascade as a separate sub-group. We identified numerous proteins involved in both the classical and lectin complement pathways but failed to identify any proteins involved in the alternative pathway. Beyond this, we found that the 21 complement associated proteins were able to separate the responder from the non-responder at any given time point by un-supervised hierarchical clustering and principal components analyses (Figure 7).

### 3.6. Classical, High Abundant Extracellular Vesical Associated Proteins

Quantitative and temporal analyses of classical, high abundant, plasma EV associated proteins revealed a dramatically less co-ordinated change in abundance (Table 6, Figure 4B). That is, very few proteins increased or decreased in at least two of the three time points analysed regardless of response to treatment. A substantial number of proteins remained generally unchanged in the patient that did not respond to treatment (~47%, 65/137) compared with the patient that did respond to treatment (~21%, 29/137) (Table 6).

## 4. Discussion

Over the last 20 years, we have made dramatic improvements in our ability to detect breast tumours early using pragmatic screening campaigns and public awareness in developed countries. However, metastatic disease remains a fatal consequence of breast cancer and occurs in approximately 30% of cases [24]. Evs. are enriched with dynamic molecular cargo that act as versatile mediators of intercellular communication. They can serve as a high-quality source of biomarkers in the research and clinical setting, as their contents are selectively packaged by parental cells. Indeed, recent reports have shown that plasma EV proteins have diagnostic utility for early breast cancer detection and prediction of recurrence risk [12]. Here, we adopted a similar study design to previous reports and used a simple protocol to enrich small Evs. < 200 nm in size (Figure 2) from serial plasma samples of three patients with hormone receptor positive, metastatic breast cancer [12,25]. A total of 470 proteins were identified and, of these, 149 were classified as EV associated proteins of interest while a further 137 were deemed representative of classical, high abundant serum EV-associated proteins. Of the 149 EV associated proteins of interest, we created a short list of 35 based on their unique abundance signature relative to response status. A review of previous studies revealed that 80% (28 out of 35) of candidate EV-associated proteins have a potential association with breast cancer aggression/metastasis/invasion and ~40% (15 out of 35) are associated with breast cancer survival (Appendix A). With the caveat that these data are generated from a small number of representative samples, the identification of such a large number of proteins associated with disease severity, metastasis and invasion supports our hypothesis that host-derived EV cargo is associated with treatment response. While it is noted that the treatment regime for each patient was somewhat different and that this could lead to the differences in the EV protein expression patterns, the intent was to characterize aggressive tumours that had different potentials for responses.

We identified five EV associated proteins that were uniquely overrepresented in all three samples from the patient who responded to therapy relative to the non-responder (Table 3). Of these proteins, previous studies have shown that high expression of glyceraldehyde-3-phosphate dehydrogenase, peroxiredoxin-2 and histone H2A type 2-A in breast cancer tissues are inversely correlated with overall survival and tumour aggressiveness [25,26,27,28]. Our own independent in silico analyses verified this observation (Appendix A). However, considering that Evs. are dynamic intercellular communication agents that functionally transfer biological information between cells, correlation between tissue and EV expression is complex in the context of functionality. In support of this, a recent prostate cancer study showed that expression of some but not all proteins was conserved between Evs. and tumour cells and that this relationship was independent of survival [29]. An absence of correlation between tissue and EV expression may indicate that certain EV packaging and processing mechanisms are altered during cancer onset and progression. Here we report high levels of EV packaged peroxiredoxin-2 in the patient with metastatic breast cancer who responded to therapy and suggest that this is uniquely associated with decreased intracellular levels. Peroxiredoxin-2 is an antioxidant enzyme that balances reactive oxygen species and cytokine-induced peroxide levels via electron transfer mediation and is considered essential for tumour cell maintenance and survival [30]. Therefore, specific EV packaging of peroxiredoxin-2 in the scenario (responder) may scavenge tumour supplies and lead to reduced cell survival, a phenomenon that is consistent with response to therapy. In other words, our data hint that EV packaged peroxiredoxin-2 may be a biologically relevant biomarker of treatment response, although further studies in larger cohorts are required to confirm this.

We observed decreased abundance of seven EV associated proteins in the non-responder at all three time points (Table 4). While some of these proteins may have potential as novel biomarkers in other cancer types, their biological relevance is also worthy of consideration. For example, solute carrier family 25 member 44 is a mitochondrial branched-chain amino acid transporter and is part of a signalling network that provides important nutrients for tumour cell proliferation [31]. Thus, reduced EV packaging of this protein is consistent with an increased demand by proliferating tumour cells. Upregulated tissue expression of Kinesin-like protein KIF20B has been reported in some solid tumours including breast, bladder and liver cancers [32] and is consistent with reduced packaging into circulating Evs. as observed in the current study. A more recent report showed that blocking cytokinesis by KIF20B inhibition increased the efficacy of microtubule-associated therapeutic agents in liver cancer [33]. Together with our findings, this suggests that a similar therapeutic advantage may be obtained in breast cancer. Serum paraoxonase/arylesterase 1 are esterase enzymes with lipophilic antioxidant characteristics. A recent report corroborating earlier studies showed that levels of circulating Paraoxonase/arylesterase 1 were lower in patients with breast cancer compared with healthy controls [34,35,36]. Our data provides further support for these findings and warrant additional targeted EV analysis of Paraoxonase/arylesterase 1 expression in larger cohorts.

In contrast, we observed increased abundance of 23 EV associated proteins in the non-responder at all three time points (Table 5). String analysis revealed the presence of two clusters within this group of proteins: (1) tubulin alpha-1B chain, cofilin-1, peptidyl-prolyl cis-trans isomerase A, pyruvate kinase PKM, fructose-bisphosphate aldolase A, fibronectin, endoplasmin, vimentin, tropomodulin-2 and collagen alpha-1(VI) chain; and (2) peregrin, histone cluster 1 H3 family member j, Ubiquitin-40S ribosomal protein S27a and histone H2B type 1-K (Supplemental Data, Figure 2). In silico analysis based on comparison with TCGA data and a published proteotranscriptomics breast cancer study [22] revealed that most of these proteins (~65%, 9 out of 14) are expressed at higher levels in breast cancer tissues compared with non-cancer controls and that 50% of these proteins are associated with shorter overall survival (Appendix A).

Many of the proteins that showed increased abundance in the non-responder are involved in tumour cell function and survival. For example, in addition to their well characterised roles in actin cytoskeleton dynamics, cofilin-1, fibronectin and fibulin-1 have also been implicated in cellular migration, tumour invasion and mitosis [37]. Consistent with our observation, cofilin showed high abundance and is associated with a lack of response to treatment, and poor outcomes for breast cancer patients [12,25,38]. In a similar context, both pyruvate kinase PKM and Fructose-bisphosphate aldolase A play important roles in glycolysis and are therefore also crucial for tumour cell proliferation. High EV levels of these proteins may reflect an increased tumoral load in patients who are unable to mount a response to treatment and highlight their potential as liquid biomarker candidates.

We were intrigued to detect an enrichment of histone associated proteins in Evs. from plasma of the non-responder. Although controversial, this is not an unusual result. Numerous other EV-proteomic studies have identified histone associated proteins enriched in cancer circulation compared with controls [12,25,39]. Whether this reflects a high histone tumoral load, deliberate shuttling and removal of these proteins or is attributed to technical factors remains unclear. Regardless, these data provide support for the notion that Evs. are implicated in DNA transcription one way or another. Evs. may be packaged with histone associated proteins in order to limit DNA accessibility and regulate transcription. On the other hand, histones may exist on the extra-vesicular periphery and may be co-incidentally detected by current isolation methods. This is a very interesting area of research that is ripe for additional discussion.

It is apparent from our proteomics analyses that plasma Evs. consist not only of tumour derived Evs. but also Evs. from different origins such as platelets, immune cells, or adipose tissues [12]. This was highlighted in our results by detection of proteins that are enriched in two complement pathways. As key mediators of inflammation, Evs. are crucial for the immune response and regulation [40]. Here, the unique deregulation of EV associated complement expression in the context of response to therapy indicates an important link with the immune response. While these data do not definitively show whether the complement proteins are intra- or inter-vesicular cargo, their coordinated deregulation in relation to response to therapy indicates their potential as biomarkers and therapeutic targets.

Beyond deliberate packaging of certain proteins as intra-vesicular cargo, we also postulate that plasma Evs. act as traps and accumulate high abundant proteins on their extra-vesicular surface while in circulation. Therefore, the overall changes in EV cargo reflect the host’s disease state in aggregate which is influenced by not only tumour cells but also the tumour microenvironment, the immune system and other physiological maintenance cascades. This is substantiated by the detection of 137 classical, high abundant serum EV-associated proteins in our analyses. Rather than consider this a limitation, we highlight these findings and show that this collection of proteins remains relatively consistent over time regardless of response to treatment. It is unrealistic to expect that Evs. isolated from plasma will recapitulate or exclusively represent the expression profile of the tumour or even of the Evs. that are released from tumour cells. However, plasma Evs. do provide a wealth of information when considered as a measure of host response.

In summary, our proof-of-concept study shows that EV associated proteins from serial blood samples offer a dynamic and versatile snapshot of the host environment. Our results hint that proteins isolated from Evs. may have roles in breast cancer metastasis and therapeutic evasion but more work is needed to realise their full translational capacity due to the limited sample size investigated here. We also highlight that Evs. offer important insight into global host dynamics beyond the tumour.

## Figures and Tables

**Figure 1 membranes-11-00880-f001:**
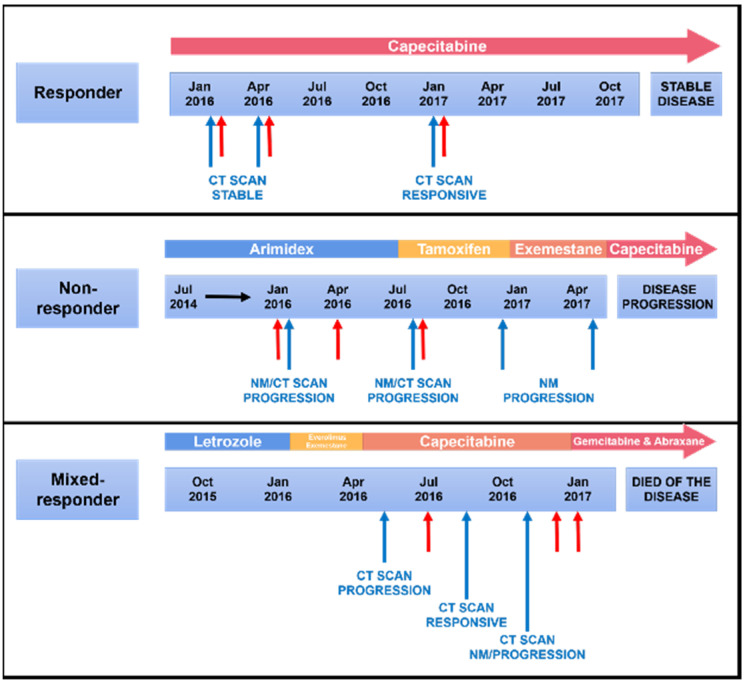
Treatment regimes, time of blood draws and response patterns. Blue and red arrows indicate time of scans and blood collection, respectively. * Computerised tomography (CT), nuclear medicine (NM).

**Figure 2 membranes-11-00880-f002:**
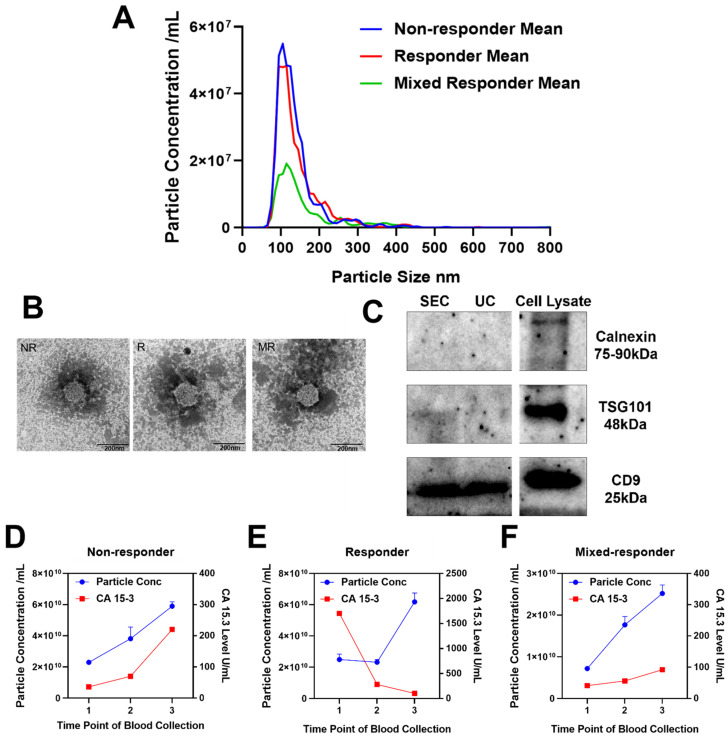
Validation of extracellular vesical isolation using nanoparticle tracking analysis, transmission electron microscopy and Western blotting. (**A**) Nanoparticle tracking analysis (NTA) was used to measure the mean size distribution and undiluted particle concentration of Evs. from three patients. (**B**) Transmission electron microscopy images of EVs. Representative images are shown for the non-responder (NR), responder (R), and mixed-responder (MR). Scale bar, 200 nm. (**C**) Representative images of Western blotting analysis from plasma Evs. (SEC pooled fractions and ultracentrifugation (UC)) and whole cell lysate of breast cancer cell line MDA-MB-231. (**D**–**F**) Line graphs were plotted to compare particle concentration and level of cancer biomarker CA 15.3.

**Figure 3 membranes-11-00880-f003:**
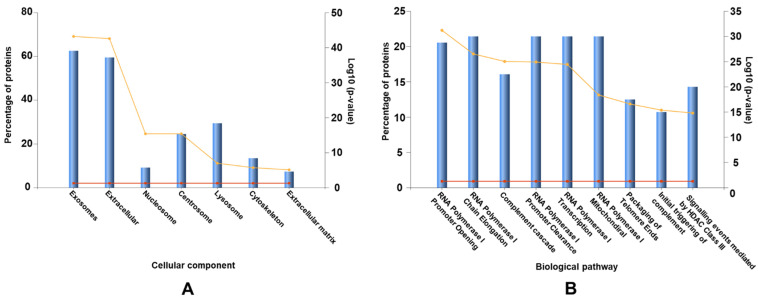
Cellular and biological pathways that are over-represented among extracellular vesicle associated proteins. EVs/exosomes were the most enriched cellular components (**A**) whereas various forms of RNA processing were the most enriched biological pathways (**B**). The x axis on the left represents the percentage of proteins (blue bars) identified in our data set that are expressed or involved in each category. The x axis on the right represents the significance of this as log 10 (*p*-value) (yellow line). The red line represents the significance at *p* = 0.05 as a reference.

**Figure 4 membranes-11-00880-f004:**
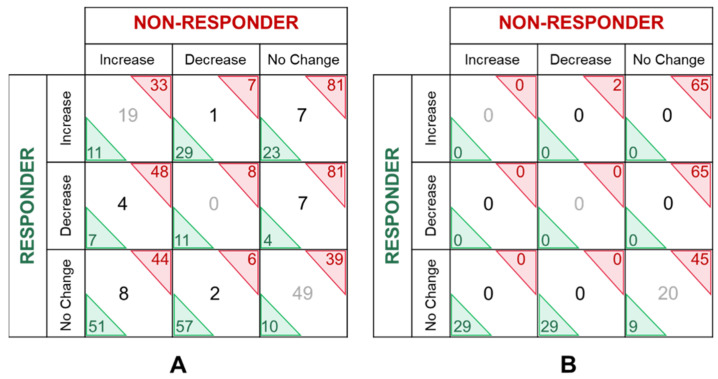
Numerical comparison of extracellular vesicle associated protein abundance changes. Proteins whose abundance was increased or decreased in at least two of the three plasma samples assayed (Table 2 and Table 6) were compared between the responder and the non-responder. Proteins were considered as candidate EV associated proteins (**A**) or classical, high abundant plasma EV associated proteins (**B**).

**Figure 5 membranes-11-00880-f005:**
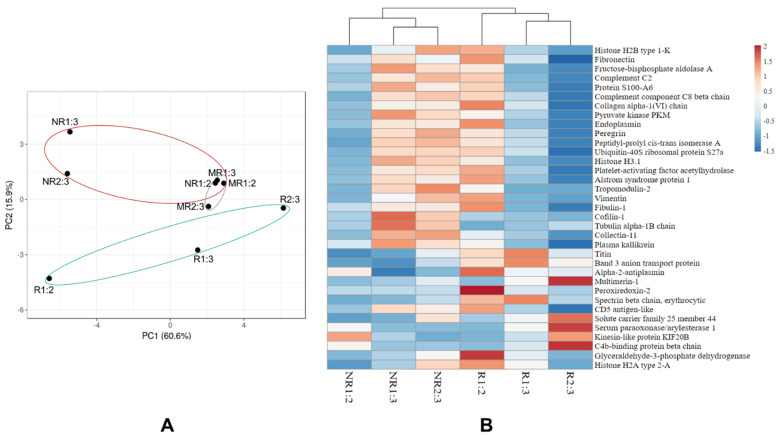
Candidate extracellular vesical associated proteins were able to discriminate the responder from the non-responder regardless of time assayed. (**A**) Unit variance scaling and SVD with imputation was used to calculate principal components. X and Y axis show principal component 1 and principal component 2 that explain 60.6% and 15.9% of the total variance, respectively. (**B**) Euclidean clustering distance and Ward clustering method were used to generate the heatmap. Temporal samples from the responder (green) or the non-responder (red) grouped closely with each other. NR, non-responder; MR, mixed-responder; R, responder.

**Figure 6 membranes-11-00880-f006:**
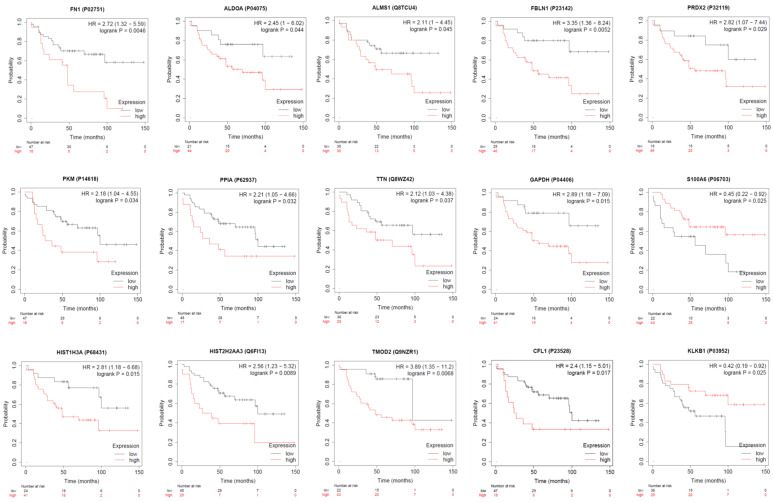
Kaplan Meier analyses showing the relationship between survival and breast tissue tumoral expression of EV associated proteins.

**Figure 7 membranes-11-00880-f007:**
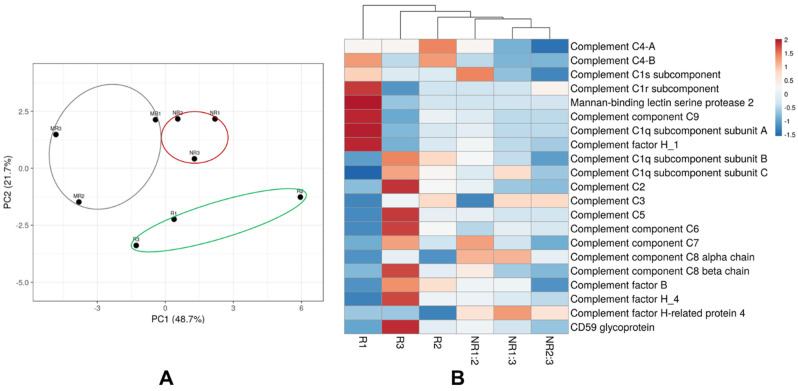
Complement associated extracellular vesical proteins were able to discriminate the responder from the non-responder regardless of time assayed. (**A**) Unit variance scaling and SVD with imputation was used to calculate principal components. X and Y axis show principal component 1 and principal component 2 that explain 48.7% and 21.7% of the total variance, respectively. (**B**) Euclidean clustering distance and Ward clustering method were used to generate the heatmap. Temporal samples from the responder (green) or the non-responder (red) grouped closely with each other.

**Table 1 membranes-11-00880-t001:** Clinical information of three breast cancer patients used to identify markers of disease using extracellular vesicles isolated from blood samples.

	Responder	Non-Responder	Mixed-Responder
Age	65	68	69
Sex	Female	Female	Female
Cancer Staging	IV	IV	IV
Histopathology	IDC, grade 2	IDC, grade 3	IDC, grade 3
Receptor status	ER positive PR positive HER2 non-amplified	ER positive PR positive HER2 non-amplified	ER 100% positive PR 10% positive HER2 non-amplified
Treatment	Endocrine treatment Chemotherapy	Endocrine treatment	Chemotherapy Radiotherapy
Response	Response at all time points	Progression at all time points	Response then progression

**Table 2 membranes-11-00880-t002:** Patterns of extracellular vesical associated proteins changed in abundance over time.

	Increase				Decrease				No Change			
Clinical Response	1–2	1–3	2–3	Overlap ≥ 2 Time Points	1–2	1–3	2–3	Overlap ≥ 2 Time Points	1–2	1–3	2–3	Overlap ≥ 2 Time Points
Non-Responder	2	55	64	**52**	12	12	5	**8**	135	82	80	**88**
Responder	61	33	34	**30**	24	14	54	**11**	64	102	61	**59**
Mixed Responder	45	38	15	**35**	60	53	30	**48**	44	58	104	**61**

**Table 3 membranes-11-00880-t003:** Extracellular vesical associated proteins that are increased in abundance in the responder.

		Non-Responder						Responder					
		Raw Abundance			Fold Change			Raw Abundance			Fold Change		
Accession	Description	NR1	NR2	NR3	NR1:2	NR1:3	NR2:3	R1	R2	R3	R1:2	R1:3	R2:3
P04406	Glyceraldehyde-3-phosphate dehydrogenase	226.3	88.3	184.8	0.390	0.817	2.093	31.6	147.9	48.9	4.680	1.547	0.331
Q6FI13	Histone H2A type 2-A	153.6	48.9	187	0.318	1.217	3.824	30.2	140.4	83.4	4.649	2.762	0.594
Q13201	Multimerin-1	132	57.1	95.3	0.433	0.722	1.669	92	38.4	215.9	0.417	2.347	5.622
P32119	Peroxiredoxin-2	122.4	93.9	89.9	0.767	0.734	0.957	10.4	150.5	18.6	14.471	1.788	0.124
P11277	Spectrin beta chain	198.9	108.9	142.8	0.548	0.718	1.311	34.2	123.2	143.2	3.602	4.187	1.162

**Table 4 membranes-11-00880-t004:** Extracellular vesical associated proteins that are decreased in abundance in the non-responder.

		Non-Responder						Responder					
		Raw Abundance			Fold Change			Raw Abundance			Fold Change		
Accession	Description	NR1	NR2	NR3	NR1:2	NR1:3	NR2:3	R1	R2	R3	R1:2	R1:3	R2:3
Q96H78	Solute carrier family 25 member 44	299.4	90.8	115	0.303	0.384	1.267	82.7	44.9	78.7	0.543	0.952	1.753
P27169	Paraoxonase/arylesterase 1	141.7	141.3	87	0.997	0.614	0.616	95.5	58	96.1	0.607	1.006	1.657
Q96Q89	Kinesin-like protein KIF20B	101.5	232	61.5	2.286	0.606	0.265	205	65.8	155.2	0.321	0.757	2.359
P20851	C4b-binding protein beta chain	136.3	130.5	54.2	0.957	0.398	0.415	150.7	53.4	101.5	0.354	0.674	1.901
Q8WZ42	Titin	151.8	66.7	94.1	0.439	0.620	1.411	54.1	113.5	141.4	2.098	2.614	1.246
P02730	Band 3 anion transport protein	175.5	90.6	76.3	0.516	0.435	0.842	59.6	88.1	109.9	1.478	1.844	1.247
P08697	Alpha-2-antiplasmin	135.4	114.5	71.3	0.846	0.527	0.623	119.8	125.4	93.5	1.047	0.780	0.746

**Table 5 membranes-11-00880-t005:** Extracellular vesical associated proteins that are increased in abundance in the non-responder.

		Non-Responder						Responder					
		Raw Abundance			Fold Change			Raw Abundance			Fold Change		
Accession	Description	NR1	NR2	NR3	NR1:2	NR1:3	NR2:3	R1	R2	R3	R1:2	R1:3	R2:3
O43866	CD5 antigen-like	69.1	79.9	184.8	1.156	2.674	2.313	72.5	231.2	94.6	3.189	1.305	0.409
O60814	Histone H2B type 1-K	125.8	63	323.2	0.501	2.569	5.130	34.5	172.4	47.7	4.997	1.383	0.277
P02751	Fibronectin	64.3	80.5	126.2	1.252	1.963	1.568	60.9	139.6	78.4	2.292	1.287	0.562
P04075	Fructose-bisphosphate aldolase A	64.6	81.3	266.4	1.259	4.124	3.277	85.2	259.5	74.5	3.046	0.874	0.287
P06681	Complement C2	95.8	78.9	189.8	0.824	1.981	2.406	88.5	200.1	64.6	2.261	0.730	0.323
P06703	Protein S100-A6	82.8	111	291.1	1.341	3.516	2.623	58.3	183.3	63.5	3.144	1.089	0.346
P07358	Complement component C8 beta chain	58.4	54.4	128.9	0.932	2.207	2.369	122.9	280.1	154.8	2.279	1.260	0.553
P12109	Collagen alpha-1(VI) chain	64	61.1	104.5	0.955	1.633	1.710	95.6	217.3	114.1	2.273	1.194	0.525
P14618	Pyruvate kinase PKM	60.9	72.4	212.1	1.189	3.483	2.930	64.5	156.1	91	2.420	1.411	0.583
P14625	Endoplasmin	85.8	89.8	223.6	1.047	2.606	2.490	53.7	189.7	56.1	3.533	1.045	0.296
P55201	Peregrin	2.5	2.2	4.8	0.880	1.920	2.182	3.5	6.4	4.0	1.829	1.143	0.625
P62937	Peptidyl-prolyl cis-trans isomerase A	81	75.5	235.3	0.932	2.905	3.117	60.3	156.9	89.9	2.602	1.491	0.573
P62979	Ubiquitin-40S ribosomal protein S27a	68	66.1	124.2	0.972	1.826	1.879	92.5	174.3	110.4	1.884	1.194	0.633
P68431	Histone H3.1	54.2	60.9	263.4	1.124	4.860	4.325	48.6	174.5	64.3	3.591	1.323	0.368
Q13093	Platelet-activating factor acetylhydrolase	97.6	89.6	189.4	0.918	1.941	2.114	89	215.2	93.1	2.418	1.046	0.433
Q8TCU4	Alstrom syndrome protein 1	72.4	82.1	235.4	1.134	3.251	2.867	72.2	282.6	85.8	3.914	1.188	0.304
Q9NZR1	Tropomodulin-2	63.9	49.3	604.3	0.772	9.457	12.258	27.3	175.8	10.6	6.440	0.388	0.060
P08670	Vimentin	101.5	59.9	316.7	0.590	3.120	5.287	31.8	184.9	19.2	5.814	0.604	0.104
P23142	Fibulin-1	115.9	115.9	204.1	1.000	1.761	1.761	52.7	127.1	35.0	2.412	0.664	0.275
P23528	Cofilin-1	41.8	54	229.9	1.292	5.500	4.257	71.7	101.7	81.9	1.418	1.142	0.805
P68363	Tubulin alpha-1B chain	58.1	72.5	252.9	1.248	4.353	3.488	100.2	76.5	105.4	0.763	1.052	1.378
Q9BWP8	Collectin-11	85.2	82.1	193.5	0.964	2.271	2.357	91	132.9	97.9	1.460	1.076	0.737

**Table 6 membranes-11-00880-t006:** Patterns of classical, high abundant extracellular vesical associated proteins that changed in abundance over time.

	Increase				Decrease				No Change			
Clinical Response	1–2	1–3	2–3	Overlap ≥ 2 Time Points	1–2	1–3	2–3	Overlap ≥ 2 Time Points	1–2	1–3	2–3	Overlap ≥ 2 Time Points
Non-Responder	11	20	16	**0**	6	37	35	**2**	120	80	86	**65**
Responder	47	30	40	**0**	32	17	48	**0**	58	90	49	**29**
Mixed-Responder	33	50	24	**2**	47	29	7	**5**	57	58	106	**11**

## Data Availability

The data were deposited in the Mass Spectrometry Interactive Virtual Environment (MassIVE) database with the dataset identifier: MSV000087527 and are publicly accessible by the following link: (https://massive.ucsd.edu/ProteoSAFe/dataset.jsp?task=4e6ad1a618d24d79b648d72af5a9e0d3).

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
