# Peer review of "Dynamic Landscape of Extracellular Vesicle-Associated Proteins Is Related to Treatment Response of Patients with Metastatic Breast Cancer"

_membranes, 2021, doi:10.3390/membranes11110880_

Round 1

Reviewer 1 Report

An impressive and well conducted original paper focuses on proteomics of extracellular plasma vesicles of 3 patients suffered from breast cancer at the same stage of disease, but differently reactive for anticancer therapy. The authors observed the differences of EVs plasma proteomic analysis between responder and non-responder for therapy  justifying to suggest the eventual protein patterns of non-responder for breast cancer therapy in comparison to the responder patient.

remarks // comments of the Reviewer

please correct line 58  - delete dash (ar-rest)                                                                                   

please correct reference No 18 (Vol & page, or article No)

Author Response

Thank you for stating that our work was “an impressive and well conducted” study. We have addressed your comments in full as below:

  1. Please correct line 58 - delete dash (ar-rest)  

Thank you for highlighting this typographical error. We have deleted the dash in arrest.                                                                       

  1. Please correct reference No 18 (Vol & page, or article No)

Thank you for alerting us to this omission. We have added volume and page number as suggested.

Reviewer 2 Report

In the current study, authors have explored the EV proteome of breast cancer patients’ blood samples collected at three different time points. The hypothesis and design of the study is appreciable and the methods are described in a detailed manner. I am worried about the sample size, which is one for each case (responder, non-responder, and mix response). Why didn’t the authors try to increase the number of patients in this study?

There are many other concerns too. For example:

  • In figure 2A, the authors mention the numbers in the range of 3.1x1010, while the y axis has values in nx107 Please confirm if it is true?
  • Also, how do authors interpret the differences in the number of EVs in three patients? Mixed response patients’ have the lowest number of EVs. Is there a possible correlation with the number and response too? The authors didn’t discuss anything about the differences in numbers among the three groups.
  • The TEM image shows just one particle, which is also not very convincing based on its morphology (looks like it is attached with broken piece of plasma membrane). Based on my prior experience, I have detected TSG101 and CD9 in my several MS runs, so the authors not detecting these proteins in their MS methods is worrisome. Overall, the quality of isolation is not convincing and needs a better TEM image and data.

The mean values of EVs in figure 2A for nonresponders are 0.6x108, while it is approximately 10 times higher in figure 2D. How come this is possible. The same is true for figures 2E and 2F.

In TMT experiment, each sample has just one replicate. This is not an ideal setup of TMT experiments for proper statistical analysis. Also, how did the authors managed to normalize the amount of peptides labeled and loaded across different TMT channels?

In line 286, ‘a particular time point with another time point’ is confusing. Please clearly state what values have been used to calculate the ratios. is it time point 2 and 3 divided by 1 or something else?

GO enriched terms also indicate that probably the isolation of EVs is not up to the mark. A lots of contaminating proteins are present, which is possibly an indicator of impurities.

Based on these observations, I believe that the results presented in the manuscript are not convincing and may be misleading. Authors need to work more on the purification part and then on the standardization of TMT experiment, by increasing the number on patients samples. They may use float channels for making comparisons across different TMTs, if needed.

Author Response

Thank you for appreciating “the hypothesis and design of the study” and highlighting that the methods were described in a detailed manner. We have addressed your comments and concerns in full as below to the best of our ability:

  1. I am worried about the sample size, which is one for each case (responder, non-responder, and mix response). Why didn’t the authors try to increase the number of patients in this study?

We agree that the sample size is small but highlight that we have multiple samples from each patient. We have described our work as a proof of concept study in an attempt to not over-interpret the data.

  1. In figure 2A, the authors mention the numbers in the range of 3.1x1010, while the y axis has values in nx107 Please confirm if it is true?

Thank you for this important comment. The difference is due to the dilution of EVs prior to NTA analyses. The representative NTA traces show undiluted data in the order of 10 to the power of 7. Once we take the dilution factor (1000x) into account, the scale changes to 10 to the power of 10. We have modified the figure legend and updated the methods to better reflect this.

  1. How do authors interpret the differences in the number of EVs in three patients? Mixed response patients’ have the lowest number of EVs. Is there a possible correlation with the number and response too? The authors didn’t discuss anything about the differences in numbers among the three groups.

Thank you for highlighting this result. Since the responder and non-responder had similar numbers of EVs, we assumed that the crude number of EVs did not correlate with response. The low number of EVs observed in the plasma of the mixed responder is difficult to interpret in the context of response to treatment. Considering the small cohort, we did not think it was helpful to elaborate on this result at this time.

  1. The TEM image shows just one particle, which is also not very convincing based on its morphology (looks like it is attached with broken piece of plasma membrane). Overall, the quality of isolation is not convincing and needs a better TEM image and data.

We agree and apologize for the poor TEM result. We have replaced this with better representative images from this experiment.

  1. Based on my prior experience, I have detected TSG101 and CD9 in my several MS runs, so the authors not detecting these proteins in their MS methods is worrisome.

We agree and are disappointed that we did not detect CD9 and TSG101 by MS. However, we did detect CD9 by western blot. Further, we detected an alternative well-defined EV protein (Heat shock protein HSP 90-beta) by MS. We hope this tempers your concerns a little. We have been as transparent as possible with the results and have strived not over interpreted the data.

  1. The mean values of EVs in figure 2A for non-responders are 0.6x108, while it is approximately 10 times higher in figure 2D. How come this is possible. The same is true for figures 2E and 2F.

Thank you for highlighting this anomaly and we are sorry about this confusing data. We attribute the differences between figure A and D-F to the dilution factor and our error when plotting the data using GraphPad. The y-axis in figures D-F were incorrect and should have represented data that had been corrected for the dilution factor. The results should have been 10 to the power of 10 (not to the power of 8). We have revised the y axis in figures D-F accordingly. The NTA traces in figure 2A are representative of time point 3 for each patient and were generated using raw, undiluted data. As such, the results in figure 2A correspond with the time point 3 results in figures D-F (when the axis is corrected). Again, we are very sorry for this confusion and our error.

  1. In TMT experiment, each sample has just one replicate. This is not an ideal setup of TMT experiments for proper statistical analysis. Also, how did the authors managed to normalize the amount of peptides labeled and loaded across different TMT channels?

We agree that the setup was not ideal. We were constrained by limited plasma that was available from the metastatic breast cancer patients. We would have performed multiple replicates if plasma was more readily available. We used the PierceTM quantitative fluorometric peptide assay (Thermo Fisher Scientific) to normalize across channels (line 154).

  1. In line 286, ‘a particular time point with another time point’ is confusing. Please clearly state what values have been used to calculate the ratios. is it time point 2 and 3 divided by 1 or something else?

We apologize for this confusing statement. We have revised the manuscript based on your helpful suggestion.

  1. GO enriched terms also indicate that probably the isolation of EVs is not up to the mark. A lots of contaminating proteins are present, which is possibly an indicator of impurities.

We appreciate this comment, thank you. We have been as transparent as possible regarding this issue. We have acknowledged that we identified extra-vesicular proteins as well as proteins that have co-isolated with EVs. We have gone as far as discussing these apparent contaminants in detail to highlight their presence in host-derived EVs. We have postulated that host-derived EVs are like ‘traps’ and used our results support this notion.

  1. Based on these observations, I believe that the results presented in the manuscript are not convincing and may be misleading.

We hope that our responses to each comment have relieved your concerns. The results are robust and reliable and we have been careful not to over interpret based on the proof of concept nature of the study.

  1. Authors need to work more on the purification part and then on the standardization of TMT experiment, by increasing the number on patients samples. They may use float channels for making comparisons across different TMTs, if needed.

These are very helpful suggestions that we will implement in future studies. We will strive to obtain more plasma from a larger cohort and will optimize purification, TMT and other elements as suggested. Unfortunately, the plasma for this study was completely expended during this project and we are unable to act accordingly at this time.

Reviewer 3 Report

Dynamic landscape of extracellular vesicle-associated proteins is related to treatment response of patients with metastatic breast cancer by Olivia Ruhen and colleagues,

In this work, the authors propose a proof of concept that can be used to identify proteins which have potential as predictors of metastatic breast cancer response to treatment.

The problem is, that It is a research work with an interesting objective but it is based on numbers and non-significant samples. To be able to resubmit this work, it is necessary to increase the number of patients. this work at present is very primitive and cannot be accepted for further review

Author Response

Thank you for reviewing our article.

We agree that the sample size is small but highlight that we have multiple samples from each patient. We have described our work as a proof of concept study in an attempt to not over-interpret the data.

Reviewer 4 Report

The authors claim to have carried out proof of concept approach to identify proteins in EVs which they claim are potential predictors of metastatic breast cancer response to treatment.

However, the authors have not convincingly demonstrated that their EV isolation method is effectively enriching EVs from serum.

NTA results are not convincing. Authors need to show NTA on the starting material (unprocessed serum) as well as EV prep to demonstrate difference.

WB is not convincing. Cell lysates, SEC and UC should be run side-by-side. Full blot should be shown in the supplemental materials. Authors need to show WB on the starting material (unprocessed serum) as well as EV prep to demonstrate EV enrichment.

While calnexin is a good negative control for cell-derived EVs, it is not a negative control for serum-derived EVs since it is not present in the serum to begin with. Albumin, IgGs and other abundant proteins are negative controls for serum. Authors need to convincingly show depletion of these negative controls instead.

EM is not convincing. The authors only showed one EV which is not enough. They need to repeat in starting material (unprocessed serum) vs EV prep to demonstrate differences.

Proteomics results are difficult to interpret. It is unknown the sequences of the peptides identified and each of their scores. Relative intensities of each protein or peptide is not shown compared to starting material (serum); e.g., how much has each protein been enriched or depleted in each EV prep relative to the starting serum. The reviewer was unable to download the proteomics results from MassIVE.

Analysis of only three clinical patients is not acceptable. Also healthy individuals are missing from the study. The authors need to do power analysis to demonstrate how many patients are needed per group. With clinical samples, they likely need to demonstrate results in at least 10 or ideally >25 patients per group.

Author Response

Please see word document for details response.

Reviewer 5 Report

The study explored the extracellular vesicle-associated proteins in relation to the treatment response of patients with metastatic breast cancer. The study was well planned with a considerable no of patients for analysis.

Please add the TEM and WB of Responder, Non-Responder, and Mixed-Responder.

Figure legend should have info about experiments, not the results. Please revise it in figure 2.

Please add the statistical analysis details in M&M.

Please clarify why particle no in Mixed-Responder is lower than the other two groups? Please includle the statistical analysis, p-value.

Why don’t you analyze the acquired data with clinicopathological parameters?

Please add the Kaplan-Meier analysis data to the main manuscript from supplementary.

Author Response

Thank you for acknowledging, “the study was well planned”. We have addressed your comments in full as below:

  1. Please add the TEM and WB of Responder, Non-Responder, and Mixed-Responder.

We have included the TEM data for each patient as requested. However as we had limited plasma from each patient for this study, we selected a random sample (non-responder) for WB validation.  Unfortunately, we do not have representative WB results for every patient. This is because we reserved as much material for MS analysis as possible.

  1. Figure legend should have info about experiments, not the results. Please revise it in figure 2.

We apologize for this oversight. We have revised the legend for figure 2 accordingly.

  1. Please add the statistical analysis details in M&M.

Due to the small number of patients and the limited sample available, statistical analyses was not performed. Therefore, we were unable to add this to the material and methods section.

  1. Please clarify why particle no in Mixed-Responder is lower than the other two groups? Please include the statistical analysis, p-value.

As the responder and the non-responder displayed similar numbers of EVs, we assumed that there was no correlation between treatment response and crude EV levels. We are perplexed about the low number of EVs in plasma from the mixed responder and are not confident that our data offers a reasonable explanation for this phenomenon. Unfortunately, we did not have sufficient replicates to perform statistical analyses and are unable to provide a p-value.

  1. Why don’t you analyze the acquired data with clinicopathological parameters?

Thank you for this question. We have not analyzed the acquired data with clinicopathological parameters due to the small sample size and in an effort to not over interpret the findings of this proof of concept study.

  1. Please add the Kaplan-Meier analysis data to the main manuscript from supplementary.

Thank you for this helpful suggestion. We have added the Kapla-Meier analysis data to the main manuscript as suggested. 

Round 2

Reviewer 2 Report

I appreciate the efforts made by authors to improve the quality of the manuscript and am happy to see the improvement in the writing and presentation both. However, my concern about the limited number of samples and some basic flaws in sample preparation and data interpretation still stands the same. I understand and duly acknowledge the difficulties in increasing the sample size. Unfortunately, in my opinion, this much of data does not seem adequate enough to draw any conclusion and make interpretations.

Author Response

We appreciate the time taken to review our manuscript a second time. We have attempted to reinforce that this is a proof of concept study due to the limited sample size.

Reviewer 4 Report

The authors have added full western blots in the supplementary data and responded to some of the comments. However, important controls are still missing to demonstrate that the authors are 'purifying' EVs. It is unclear whether EV isolation is even necessary or can all these markers be detected in the unprocessed serum. Here are the specific comments:
-Authors have provided technical reasoning for why unprocessed serum could not be used for NTA and TEM. However, Western blots of unprocessed and processed serum is necessary. Equal amounts of protein should be loaded per lane for this comparison to avoid distortion of protein bands. Enrichment of positive controls (CD9 and TSG101) and depletion of negative control (Calnexin) is expected.
-Authors explanation of using calnexin as a negative control is acceptable. However, its presence in unprocessed serum should be demonstrated by WB above to confirm it is depleted.
-List of peptides and their scores should be included in the supplemental materials. The reviewer was still unable to access the reported files from the link and other readers could have similar issues. 

Author Response

Thank you very much for re-reviewing our manuscript. We value your time and comments.

We have responded to each specific comment below:

1. ‘Western blots of unprocessed and processed serum is necessary. Equal amounts of protein should be loaded per lane for this comparison to avoid distortion of protein bands. Enrichment of positive controls (CD9 and TSG101) and depletion of negative control (Calnexin) is expected.’ We would like to reiterate that we have performed Western Blot characterisation of EVs according to internationally accepted guidelines, which notes that ‘this comparison [of EVs and source material] can be easily performed only for analysis of EVs from cell culture conditioned medium; it is more difficult for biofluids (in which EVs may originate from cells in the fluid, but also from cells delimiting the fluid canals, and thus are difficult to attribute in bulk to any given cell type)’ (1). This is supported in recent published examples of studies in which proteomic analysis was performed on EV samples derived from serum/plasma in a similar manner to our own study, which do not include comparison of the source serum/plasma to EV samples for EV characterisation (2,3,4). Furthermore, it would not be possible to run processed (i.e. EV-depleted) serum/plasma as this cannot be recovered from the size-exclusion chromatography column after being used for EV isolation.

2. ‘Authors explanation of using calnexin as a negative control is acceptable. However, its presence in unprocessed serum should be demonstrated by WB above to confirm it is depleted.’ As calnexin is an intracellular protein, cell lysate from a breast cancer cell line (MDA-MB-231) was included in our Western blot to demonstrate absence of cellular material in our EV preparations in line with Minimal Information for Studies of Extracellular Vesicles (MISEV) recommendations (1).

3. Access to raw data/supplementary materials: We are sorry for all the issues with regard to accessing this data. We are confident that the full list of identified peptides and respective scores is accessible via MassIVE but now realize that a dedicated, yet freely available FTP client is required. We recently learnt that Chrome and other major browsers have dropped access to FTP and we believe that is why accessing the data was problematic. Therefore to access the data, the reviewers need to download WinSCP or a similar FTP client application as per the instructions on massIVE. Then, please use the following:

                            Host: massive.ucsd.edu

                            Username: MSV000087527

                            Password: a

Reference list:

  1. Théry, C. et al. Minimal information for studies of extracellular vesicles 2018 (MISEV2018): a position statement of the International Society for Extracellular Vesicles and update of the MISEV2014 guidelines. J Extracell Vesicles 2018, 7, 1535750, doi:10.1080/20013078.2018.1535750.
  2. Ni, H. et al. Label-free proteomic analysis of serum exosomes from paroxysmal atrial fibrillation patients. Clin Proteom 18, 1 (2021). https://doi.org/10.1186/s12014-020-09304-8
  3. Anastasi, F. et al. Proteomics analysis of serum small extracellular vesicles for the longitudinal study of a glioblastoma multiforme mouse model. Sci Rep 10, 20498 (2020). https://doi.org/10.1038/s41598-020-77535-8
  4. Wei, R., Zhao, L., Kong, G. et al. Combination of Size-Exclusion Chromatography and Ultracentrifugation Improves the Proteomic Profiling of Plasma-Derived Small Extracellular Vesicles. Biol Proced Online 22, 12 (2020). https://doi.org/10.1186/s12575-020-00125-5

Reviewer 5 Report

Since they used only a few samples for analysis, the TEM and WB are from different samples. Lacks Scientific Soundnes.

Lacks in statistical analysis. 

Author Response

We appreciate the time taken to review our manuscript a second time.